# A CMOS Hybrid System for Non-Invasive Hemoglobin and Oxygen Saturation Monitoring with Super Wavelength Infrared Light Emitting Diodes

**DOI:** 10.3390/mi16101086

**Published:** 2025-09-25

**Authors:** Hyunjin Park, Seoyeon Kang, Jiwon Kim, Jeena Lee, Somi Park, Sung-Min Park

**Affiliations:** Division of Electronic and Semiconductor Engineering, Ewha Womans University, Seoul 03760, Republic of Korea; hyunjinpark@ewha.ac.kr (H.P.); kiki21ee@ewha.ac.kr (S.K.); kkjjww9018@ewhain.net (J.K.); dlwlsk@ewha.ac.kr (J.L.)

**Keywords:** heart rate, hemoglobin, multi-wavelength sensing, noninvasive, oxygen saturation, photoplethysmography

## Abstract

This paper presents a CMOS-based hybrid system capable of noninvasively quantifying the total hemoglobin (tHb), the oxygen saturation (SpO_2_), and the heart rate (HR) by utilizing five-wavelength (670, 770, 810, 850, and 950 nm) photoplethysmography. Conventional pulse oximeters are limited to the measurements of SpO_2_ and heart rate, therefore hindering the real-time estimation of tHb that is clinically essential for monitoring anemia, chronic diseases, and postoperative recovery. Therefore, the proposed hybrid system enables us to distinguish between the concentrations of oxygenated (HbO_2_) and deoxygenated hemoglobin (Hb) by using the absorption characteristics of five wavelengths from the visible to near-infrared range. This CMOS hybrid mixed-signal architecture includes a light emitting diode (LED) driver as a transmitter and an optoelectronic receiver with on-chip avalanche photodiodes, followed by a field-programmable gate array (FPGA) for a real-time signal processing pipeline. The proposed hybrid system, validated through post-layout simulations and algorithmic verification, achieves high precision with ±0.3 g/dL accuracy for tHb and ±1.5% for SpO_2_, while the heart rate is extracted via 1024-point Fast Fourier Transform (FFT) with an error below ±0.2%. These results demonstrate the potential of a CMOS-based hybrid system as a feasible solution to achieve real-time, low-power, and high-accuracy analysis of bio-signals for clinical and home-use applications.

## 1. Introduction

This paper proposes a CMOS-based hybrid measurement system for bio-signals that is capable of quantifying the total hemoglobin (tHb), the blood oxygen saturation (SpO_2_), and the heart rate (HR) non-invasively and simultaneously. Conventional pulse oximeters are restricted to the monitoring of SpO_2_ and heart rate only. Therefore, they lack the ability to measure tHb, which is essential for assessing oxygen transport capacity and for managing conditions such as anemia, chronic disease, and postoperative recovery. Monitoring both tHb and oxyhemoglobin (HbO_2_) levels is particularly critical to assess the body’s oxygen transport capacity and is essential for patients with anemia, chronic diseases, or those recovering from surgery. Table 1 defines tHb andSpO_2_ with their normal range.

The proposed CMOS-based hybrid system utilizes photoplethysmography (PPG) technology, which captures the periodic changes in blood volume associated with the cardiac cycle [1]. This leads to variations in the amounts of absorbed lights as it passes through or is reflected by tissues. These fluctuations result in electrical signals that contain rich physiological information [2,3]. However, a light source and a photodetector are required to obtain PPG signals that consist of AC and DC components (Figure 1a) [2,4]. While the AC component represents the pulsatile changes in arterial blood due to variations in blood volume, the DC component arises from non-pulsatile components such as venous blood and other tissues. In consequence, as light passes through or is reflected by the tissue, these volumetric changes cause variations in the amount of light absorbed. These fluctuations generate electrical signals that contain rich physiological information.

HbO_2_ and Hb exhibit distinct light absorption characteristics at different wavelengths. Figure 1b shows that 670 nm lies near an absorption valley of HbO_2_ and thus accentuates Hb absorption, whereas 950 nm exhibits the opposite trend (greater relative HbO_2_ absorption) [5]. Therefore, this wavelength-dependent difference in absorption enables the separation and quantification of blood components. In this work, five wavelengths (670, 770, 810, 850, and 950 nm) are carefully selected according to the absorption spectra of HbO_2_ and Hb reported in the literature. In particular, 810 nm is chosen as the isosbestic point where both species absorb equally, providing a reference wavelength independent of oxygen saturation. To further enhance accuracy, two additional wavelengths are employed at either side of the isosbestic point: 770 nm (left of 810 nm, where Hb absorption dominates) and 850 nm (right of 810 nm, where HbO_2_ absorption dominates). By adopting this multi-wavelength PPG approach, we can reduce the measurement error relative to the three-wavelength method (as detailed in Section 2). Based on the Beer–Lambert law, the quantification of tHb concentration is achieved by utilizing the ratio of AC to DC components at each wavelength. In simulations, the proposed method achieves a precision of ±0.3 g/dL for tHb compared with blood tests, an accuracy of ±1.5% for SpO_2_ against commercial pulse oximeters, and an accuracy of ±0.2% for HR relative to MATLAB R2023a reference values. Table 2 compares the absorption rate of HbO_2_ and Hb with respect to wavelengths.

## 2. Related Previous Works

While non-invasive bio-signal measurement technology is continuously advancing, existing studies have shown limitations in terms of the variety of measurable parameters and robustness against disturbances. This section reviews the literature, from commercialized technologies to alternative approaches, to clarify the necessity of the simultaneous quantification of HR, SpO_2_, and tHb based on the five-wavelength PPG proposed in this paper.

Commercial pulse oximeters (POs) are typically based on PPG technology that uses a dual-wavelength light source, most commonly red (approximately 660 nm) and infrared (IR, approximately 940 nm) [6]. The calculation method for this dual-wavelength approach is summarized as a ratio-metric analysis based on the Beer–Lambert law. As described in (1), the absorbance of a mixture is the sum of the absorbances of its components.(1)$$A_{\lambda }=\epsilon_{\lambda 1}\times C_{1}\times L+\epsilon_{\lambda 2}\times C_{2}\times L,$$
where $C_{1}$ and $C_{2}$ are the unknown concentrations of components 1 and 2 (e.g., HbO_2_ and Hb), $\epsilon_{\lambda 1}$ and $\epsilon_{\lambda 2}$ are the extinction coefficients of components 1 and 2 for a given wavelength, and L is the optical path length [3].

For the pulsatile component of the signal, the change in absorbance, $\Delta A_{\lambda }$, is proportional to the change in concentration (ΔC) and the change in optical path length (ΔL). As shown in (2), the R-value is an effective method that mathematically cancels out the unknown variables such as C and ΔL in the process of dividing the AC/DC ratios of the two wavelengths [7]. Subsequently, SpO_2_ is determined through an empirical calibration function, f(⋅) [8,9], as(2)$$R=\frac{{\left(\frac{AC}{DC}\right)}_{Red}}{{\left(\frac{AC}{DC}\right)}_{IR}},\;SpO_{2}\left(\%\right)=f(R),$$
where *f*(⋅) is determined through a clinical calibration curve that compares the R-values measured by the device with the actual arterial oxygen saturation (SaO_2_) values analyzed by a gold-standard CO-oximeter [10]. However, in this process of variable cancelation, information related to the total blood volume would be lost. Consequently, while the proportion of SpO_2_ can be determined, absolute concentrations such as tHb cannot be estimated. Although estimating tHb by using only two wavelengths might be feasible, it generally relies on device-specific calibration and modeling approximations [11].

To overcome the limitations of tHb measurement, some studies have proposed alternative approaches that analyze the correlation between morphological features of the electrocardiogram (ECG) signal and hemoglobin levels [12]. However, this method requires attaching separate electrodes to the body, which increases hardware complexity compared to a single optical sensor system. Against this background, research has progressed toward increasing the number of PPG wavelengths. While a three-wavelength system could solve for two unknowns (the concentrations of HbO_2_ and Hb) by using three equations, significant measurement errors would arise if the effect of light scattering in tissue is ignored. Therefore, to achieve higher accuracy, a third unknown—the Tissue Scattering Coefficient (S), which represents the magnitude of the scattering effect—was introduced. Equation (3) represents the method for estimating tHb by forming a 3 × 3 determined system with the concentrations of HbO_2_, Hb, and the Tissue Scattering Coefficient (S) as the unknowns [13].(3)$$\left[\begin{array}{c} R_{\lambda 1}\\ R_{\lambda 2}\\ R_{\lambda 3} \end{array}\right]=\left[\begin{array}{ccc} \begin{array}{c} \epsilon_{1,Hb}\\ \epsilon_{2,Hb}\\ \epsilon_{3,Hb} \end{array} & \begin{array}{c} \epsilon_{1,HbO_{2}}\\ \epsilon_{2,HbO_{2}}\\ \epsilon_{3,HbO_{2}} \end{array} & \begin{array}{c} k_{1}\\ k_{2}\\ k_{3} \end{array} \end{array}\right]\left[\begin{array}{c} C_{Hb}\\ C_{HbO_{2}}\\ S \end{array}\right],$$
where *k* is the wavelength-dependent scattering weight that quantifies how much the light at a specific wavelength is attenuated by the overall scattering properties (S) of the tissue. It is determined either through theoretical modeling or statistically from experimental data.

Nonetheless, a determined system is highly susceptible to the subtle noise present in biological signals, which leads to the amplification of measurement errors. In contrast, the multi-wavelength method proposed in this paper constitutes an overdetermined system, as described in (4), which uses multiple measurements to estimate two unknowns, i.e., the concentrations of HbO_2_ and Hb [14,15,16]. Compared to the three-wavelength configuration, adding two more wavelengths reduces the mean absolute error from 1.377 g/dL to 0.3499 g/dL [17]. Employing the least-squares method to find an optimal solution that minimizes the overall error across all measurements enables higher measurement accuracy [18]. However, a limitation of a previous study that determines hemoglobin concentration by using five-wavelength PPG was its reliance on an embedded platform that is incapable of hardware synthesis. In contrast, this work implements a CMOS hybrid system that receives the signal via an analog front-end, converts it to a digital format for computation, and applies an integrated calculation module [17].(4)$$\left[\begin{array}{c} R_{\lambda 1}\\ R_{\lambda 2}\\ R_{\lambda 3}\\ \vdots \\ R_{\lambda n} \end{array}\right]=\left[\begin{array}{c} \begin{array}{cc} \epsilon_{1,\;Hb} & \epsilon_{1,HbO_{2}}\\ \epsilon_{2,\;Hb} & \epsilon_{2,HbO_{2}}\\ \epsilon_{3,\;Hb} & \epsilon_{3,HbO_{2}} \end{array}\\ \begin{array}{cc} \vdots & \vdots \\ \epsilon_{n,Hb} & \epsilon_{n,HbO_{2}} \end{array} \end{array}\right]\left[\begin{array}{c} C_{Hb}\\ C_{HbO_{2}} \end{array}\right]$$

## 3. Circuit Description

In this section, a stand-alone optical monitoring system for bio-signals is designed for accurate and noninvasive measurements of the tHb, the SpO_2_, and the heart rate in real time. For this purpose, the proposed system consists of a current-mode LED driver, a high-gain transimpedance amplifier (TIA), an analog filter module, and a digital signal processing algorithm into a unified transmit–receive–compute architecture where the photoplethysmography (PPG) signals can be optically acquired and quantitatively processed to extract the information of the tHb concentration, the SpO_2_, and the pulse rate. Also, the proposed monitoring system targets the following quantitative performance (as described in Table 3) to ensure reliable operations in clinical and mobile healthcare environments, such as hospitals, home care settings, and emergency medical sites. Its low-power and high-precision characteristics render it particularly suitable for resource-constrained or rapidly deployable applications, including mobile clinics or disaster response units.

Figure 2 illustrates the system-level architecture of the proposed hybrid multi-wavelength PPG monitoring system. The LED driver as a transmitter (Tx) sequentially activates the five LEDs, in which each LED emits at a different wavelength to illuminate the tissue. The transmitted optical signals are captured by an on-chip avalanche photodiode (APD) in the receiver (Rx) that is an integrated high-gain transimpedance amplifier (TIA). Then, the analog signal is digitized into a 20-bit digital code by using an FPGA-based analog-to-digital converter (ADC). In particular, a digital low-pass filter (with 3 Hz bandwidth) is applied to suppress high-frequency noises. In consequence, the resulting signals are processed to extract the SpO_2_ and tHb concentration.

### 3.1. Analog Circuit Design

Figure 3 shows the chip layout of the analog front-end circuits, including the LED driver (Tx) and the optoelectronic TIA (Rx), which were integrated within a 1 × 1 mm^2^ die by using standard 180 nm CMOS technology. The core circuit of the LED driver occupies an area of 230 × 168 µm^2^, while that of the optoelectronic Rx occupies an area of 240 × 160 µm^2^, respectively.

Figure 4 illustrates the block diagram of the proposed LED driver and the equivalent circuit of an LED device. This driver adopts a current-mode modulation architecture where a reference current (I_REF_) is generated and mirrored via a cascode current mirror circuit. In addition, NMOS switches are exploited to selectively control the reference current to form a modulation current (I_MOD_). This modulation current is then combined with a fixed bias current (I_BIAS_) to generate the total output current (up to 23.2 mA), thus ensuring reliable LED operations across all channels because each LED requires approximately 20 mA of forward current for activation [19].

Figure 5 presents the complete schematic diagram of the LED driver, comprising a startup circuit, a bias generator, and a wide-swing cascode current mirror. The reference current generator, as shown in Figure 6a, defines the transistor dimensions as M1 = M2 = (W/L)_P_, M5 = (W/L)_N_, and M6 = K* (W/L)_N_. Based on these ratios, the reference current (I_REF_) is generated as a function of the external resistor (R_S_) and the device geometry, as described in the following equation, i.e., $I_{ref}=\frac{2}{\mu_{n}C_{ox}({W/L}_{N})}\frac{1}{R_{S}^{2}}({1-\frac{1}{\sqrt{K}})}^{2}$.

Furthermore, the transconductance (gₘ) of M5 becomes independent of the process, voltage, and temperature (PVT) variations and remains constant once the parameters W/L and R_S_ are fixed, as derived in the following equation of $g_{m}=\sqrt{2\mu_{n}C_{ox}{\left(W/L\right)}_{N}I_{D1}}$.

The startup circuit employs a diode-connected transistor (M_SU_) to prevent ‘dead zone’ where I_D1_ = I_D2_ = 0 at the steady state. When the startup circuit is initially powered and the current flow is absent, the gate of M4 is pulled to V_DD_ and the gate of M1 to ground, thereby turning on the diode-connected transistor (M_SU_). This initiates the transitions of the gate voltages in M1 and M4, thus allowing the DC current to start flowing through M_SU_, M1, and M4. These startup operations are possible only when $V_{TH1}+V_{TH3}+V_{TH5}<V_{DD}$. Once the steady-state operation is reached, the gates of M1 and M4 converge, hence causing the gate-source voltage (V_GS_) of M_SU_ to fall to 0 V and finally turning off M_SU_. This final steady state is valid under the conditions of $V_{GS1}+V_{GS3}<V_{DD}$ and $V_{GS1}+V_{GS3}+V_{TH5}<V_{DD}$.

In a conventional topology (shown in Figure 6b), this issue is exacerbated under the process, voltage, and temperature (PVT) variations that affect both the threshold voltage and the resistance of R_5_. To overcome these limitations, we have employed the improved design of the wide-swing cascode current mirror (as depicted in Figure 6c) that replaces the passive resistor with an NMOS device (M5) and introduces a dedicated bias generation circuit to establish the gate voltage V_BIAS_ at M9. This modification alleviates the voltage headroom issue and enhances the robustness even against the severe PVT shifts. Therefore, it should be noted that the following conditions must be satisfied to ensure both M4 and M5 operate in the saturation region, i.e., $V_{DS4}>V_{GS4}-V_{TH4}$, $V_{DS5}>V_{GS5}-V_{TH5}$. Solving these inequalities yields the allowable range of V_BIAS_, that is, $V_{GS5}+V_{GS4}-V_{TH4}<V_{BIAS}<V_{GS4}+V_{TH5}$.

Meanwhile, the proposed LED driver includes six parallel channels (IN1–IN6), which are controlled by NMOS switches (M28–M33). Unlike conventional topologies, where each channel drives a separate LED, six channels in this architecture jointly contribute to the total modulation current that biases a single multi-chip LED emitter. Also, the driver can sequentially activate the five wavelengths (670, 770, 810, 850, and 950 nm) by switching the target LED within the package, while the combined channel current provides the required drive strength. As a result, the output current (I_OUT_) reaches 20.4 mA in the normal mode and 23.2 mA in the high-output mode, which is sufficient to ensure reliable operations of all wavelengths in the multi-chip emitter.

Figure 7 presents the transient simulation results of the LED driver, which confirms that the stable output currents can be maintained in both normal and high-power modes. Under the maximum current condition of 23.2 mA, the total power consumption becomes 73.4 mW. As reported in [20], the optical output power of the LEDs exhibits an almost linear dependence on the drive current within the range of 5–30 mA. Therefore, a 13.7% increase in the modulation current (I_MOD_) would lead to a 13% increase in the emitted optical power. These results further indicate that the 13.7% increase of I_MOD_ enhances the amplitude of the PPG AC component by 15–17% and improves the signal-to-noise ratio (SNR) by approximately 6.6% [21]. These improvements are especially beneficial for subjects with high melanin content or when measuring at peripheral regions with limited blood perfusion.

Figure 8 illustrates the transistor-level schematic diagram of the proposed optoelectronic receiver (Rx), which consists of an on-chip avalanche photodiode (APD) for photocurrent generation, a shielded dummy APD that is electrically matched to the primary APD but is shielded by a metal layer to block incident light for circuit symmetry, input resistors (R3, R4) for current-to-voltage conversion, and an instrumentation amplifier followed by a difference amplifier for signal amplification.

At the dummy-APD input node, only the common-mode voltage V_ICM_ set by the current-to-voltage conversion resistors (R3 and R4) is present, together with inevitable noises. At the on-chip APD input node, the desired LED-induced signal is superimposed on these components. To suppress the undesired common-mode signals, the optoelectronic Rx adopts a novel architecture based on instrumentation amplifier (IA) followed by a difference amplifier. This would effectively reject common-mode offsets and noises, expanding the input dynamic range and enabling robust amplification of the LED-induced signals.

The proposed optoelectronic Rx is a transimpedance amplifier (TIA) based on IA; the circuit is called IA-TIA hereafter. Then, the total transimpedance gain is given by $A_{d}=\frac{v_{out}}{i_{pd}}=\left(R_{3}||R_{4}\right)\left(1+\frac{{2R}_{6}}{R_{5}}\right)\frac{R_{11}}{R_{8}}$. In the presence of an input common-mode signal (V_ICM_), the voltage at each operational amplifier (op-amp) remains identical, and hence the output remains unaffected because the common-mode gain is given by $A_{CM}=\frac{v_{out}}{i_{icm}}=\frac{v_{o1}}{i_{icm}}\times \frac{v_{out}}{v_{o1}}=\left(R_{3}||R_{4}\right)\left(\frac{1}{R_{8}+R_{11}}\right)\left(1-\frac{R_{11}}{R_{8}}\frac{R_{8}}{R_{11}}\right)$ for the difference amplifier stage. This results in A_CM_ = 0, indicating that the common-mode rejection ratio (CMRR) can be theoretically infinite. Such performance is beneficial in suppressing the DC offsets and enhancing the signal integrity under the ambient noise conditions. The op-amps exploit a two-stage topology with differential input and single-ended output, where a series combination of a compensation capacitor (C_1_) and resistor (R_2_) is added to ensure a phase margin exceeding 60°, thereby avoiding gain peaking and ensuring frequency stability.

Meanwhile, the Miller capacitance is incorporated into the IA structure not only to improve robustness further, but also to mitigate the potential instability due to its high gain. Therefore, the complete Rx circuit functions as a high-precision amplifier that can effectively reject the DC components caused by the ambient lights.

Figure 9a shows the simulated frequency response of a conventional IA without a Miller compensation, achieving a transimpedance gain of 126 dBΩ and a bandwidth of 135 kHz. However, the resulting bandwidth is considerably broader than the physiological signal band of interest (e.g., ECG or PPG signals typically < 3 Hz), rendering the system vulnerable to high-frequency noise and environmental interference. To mitigate this issue, a modified IA-TIA architecture incorporates a Miller capacitor, in which the PMOS transistors (M10 and M12) serve as active loads to improve gain linearity and output swing. This Miller compensation technique reduces the bandwidth to 85 kHz while improving the gain to 129 dBΩ, thereby enhancing the signal integrity without compromising sensitivity, as shown in Figure 9b. The total power consumption of the IA-TIA is 19 mW.

Figure 9c illustrates the noise performance of the proposed IA-TIA, where the input-referred noise flattens at 197 pA/√Hz beyond the corner frequency. Then, a high optical sensitivity of −30.8 dBm is achieved, in which the responsivity of the APD was experimentally measured to be 2.72 A/W.

Figure 10a presents the simulated pulse response of the optoelectronic Rx (IA-TIA), where the output maintains linearity for small input currents but exhibits saturation with large input currents. Figure 10b shows the simulated eye diagrams at an operating speed of 100 kb/s, confirming wide and clean eye-opening across a wide range of input signals. Figure 11 displays the transient response for each wavelength, indicating that distinct AC and DC values can be generated as anticipated. For system-level validation, real bio-signals acquired from a multi-wavelength sensor are used as input. These distinct outputs were extracted in CSV format and processed through an FPGA, thus demonstrating the full integration of the analog front-end and digital signal processing blocks within a mixed-signal architecture.

### 3.2. Digital Circuit

The digital processing block is designed to extract physiological parameters—tHb, HbO_2_, SpO_2_, and heart rate—from the raw PPG signals received by the analog front-end. Figure 12 illustrates the overall signal flow, starting from the analog-to-digital converted (ADC) Rx output and passing through the band-pass and low-pass filtering, the FFT-based heart rate estimation, the AC/DC extraction, and the MBLL-based concentration calculation with calibration. These designs are implemented entirely by utilizing Verilog HDL and operate in a fully parallel pipeline structure to ensure real-time processing.

#### 3.2.1. The 8th Order IIR Digital Band-Pass Filter (BPF)

An 8th order infinite impulse response (IIR) band-pass filter (BPF) is applied to isolate bio-signals within the 0.5–3.0 Hz range [2]. The filter follows the structure$$y[n]=b_{0}x[n]+b_{1}x[n-1]+b2x[n-2]-a_{1}y[n-1]-a2y[n-2]$$
and is implemented in Q2.23 fixed-point format based on a 200 Hz sampling frequency. The BPF also suppresses DC components at 0 Hz in preparation for frequency-domain analysis. Figure 13 presents the data comparison before and after the filtering from the raw data to the filtered output.

#### 3.2.2. The 60-Tap FIR Digital Low-Pass Filter (LPF)

A 60-tap finite impulse response (FIR) low-pass filter (LPF) is employed to remove high-frequency noise. With a cutoff frequency of 3 Hz, the filter is also implemented in Q2.23 format and follows the convolutional form $y[n]\;=\;\sum_{k=0}^{N-1}h[k]\;x[n\;-\;k]$. The filtering results are presented in Figure 13b and Figure 13c, respectively.
Figure 13Comparison of simulated PPG data before and after filtering: (**a**) raw data (output from Rx), (**b**) data after low-pass filer (LPF), and (**c**) data after band-pass filter (BPF).
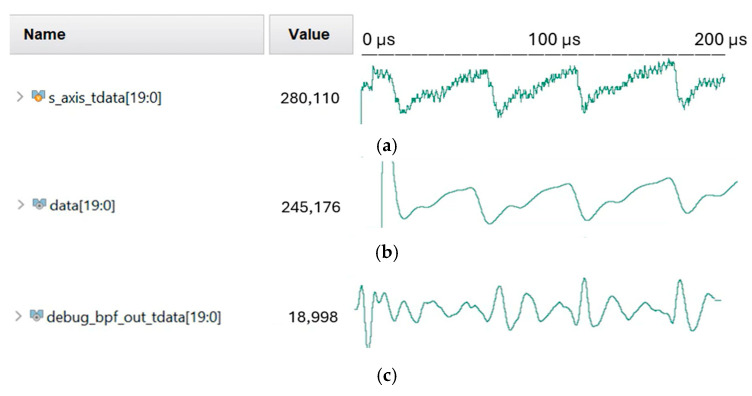


#### 3.2.3. The 1024-Point Fast Fourier Transform (FFT)

The dominant frequency component and the heart rate of the PPG signal are extracted based on the index (maxbin) corresponding to the maximum magnitude of the FFT spectrum and the sampling frequency (FS). The index count starts from 1, and the calculated frequency is separated into integer and fractional components for precise representation. The equations used for computing the dominant frequency and the heart rate (HR) are given by ${freq}_{PPG}=\frac{FS\times (maxbin-1)}{1024}$, $HeartRate\left(bpm\right)=60\times {freq}_{PPG}$. Figure 14a illustrates the procedure for identifying the maximum FFT magnitude, while Figure 14b shows the extracted maxbin index and the corresponding FFT result. The FFT is implemented as a 1024-point radix-2 Cooley–Tukey algorithm consisting of ten stages of butterfly operations. The input sequence is first reordered in bit-reversed order [22,23]. Complex multiplications in each stage are performed using precomputed twiddle factors of the form cos(2πk/N)−j·sin(2πk/N), stored in a lookup table. These twiddle factors are applied in the complex addition and subtraction between data pairs across each stage.

#### 3.2.4. Data Reader

A dedicated data management module tracks the number of samples per chunk and maintains the LED-specific chunk indices to enable the synchronized multi-channel operations.

#### 3.2.5. AC (Peak-to-Peak) Amplitude Extraction

The AC component of the PPG signal is extracted as the peak-to-peak value. Peaks are detected by using a three-point sliding window, where a point qualifies as a peak if (prev1 > prev2) && (prev1 > curr). To ensure the physiological relevance, only peaks separated by a minimum interval (Period) are considered valid. Then, the AC value is calculated as |max_ave − min_ave|, as shown in Figure 15a.

#### 3.2.6. DC (Chunk Average) Level Estimation

The DC component, representing the baseline tissue absorption, is estimated by averaging the local minimum values of the PPG signal for each wavelength segment. These minima correspond to the time points with maximal light transmission and are depicted in Figure 15b.

#### 3.2.7. Calculation (Modified Beer–Lambert Law)

The PPG leverages the wavelength-dependent absorption characteristics of HbO_2_ and Hb to estimate the blood composition. This phenomenon is modeled by the Modified Beer–Lambert Law (MBLL), which relates the light absorbance to the concentration and the optical properties of tissue. The general equation is given by$$A_{\lambda }=\epsilon_{\lambda ,Hb}\cdot [Hb]\cdot L+\epsilon_{\lambda ,HbO_{2}}\cdot [HbO_{2}]\cdot L,$$
where *A*(*λ*) is the absorbance at wavelength *λ*, *ε*(*λ*) is the molar extinction coefficient, *C* is the concentration of the absorbing species, and *d* is the optical path length.

To apply the MBLL to the PPG signals, the AC and DC components extracted from the modules of Section 3.2.5 and Section 3.2.6 are utilized to compute a normalized absorbance, yielding $A_{\lambda }=\log_{10}(\frac{{AC}_{\lambda }}{{DC}_{\lambda }})$.

Using the multi-wavelength measurements, we solve the linear system to estimate the hemoglobin concentration $C_{Hb}$
and $C_{HbO_{2}}$
, as defined in (4). Specifically, the least-squares solution is obtained by computing the pseudo-inverse of the extinction coefficient matrix, as shown below.$$\left[\begin{array}{c} Hb\\ HbO_{2} \end{array}\right]=\left({\left[\begin{array}{c} \begin{array}{cc} \epsilon_{670,\;Hb} & \epsilon_{670,HbO_{2}}\\ \epsilon_{770,\;Hb} & \epsilon_{770,HbO_{2}}\\ \epsilon_{810,\;Hb} & \epsilon_{810,HbO_{2}} \end{array}\\ \begin{array}{cc} \epsilon_{850,\;Hb} & \epsilon_{850,HbO_{2}}\\ \epsilon_{950,\;Hb} & \epsilon_{950,HbO_{2}} \end{array} \end{array}\right]}^{T}{\left[\begin{array}{c} \begin{array}{cc} \epsilon_{670,\;Hb} & \epsilon_{670,HbO_{2}}\\ \epsilon_{770,\;Hb} & \epsilon_{770,HbO_{2}}\\ \epsilon_{810,\;Hb} & \epsilon_{810,HbO_{2}} \end{array}\\ \begin{array}{cc} \epsilon_{850,\;Hb} & \epsilon_{850,HbO_{2}}\\ \epsilon_{950,\;Hb} & \epsilon_{950,HbO_{2}} \end{array} \end{array}\right]}^{-1}\right)\cdot \left[\begin{array}{c} A_{670}\\ A_{770}\\ A_{810}\\ A_{850}\\ A_{950} \end{array}\right],$$
where *ϵ* is the extinction coefficient matrix (dimensions: wavelengths × 2), and *A* is the absorbance vector across wavelengths. The resulting vector contains the estimated concentrations of HbO_2_ and Hb.

#### 3.2.8. Calibration

The Hb and HbO_2_ values calculated via MBLL are linearly calibrated by using actual reference measurements to reduce deviations. A calibration curve is constructed by utilizing three real measurement points via polynomial fitting (polyfit), and is expressed as a first-order function, where x represents the uncalibrated value and y represents the calibrated (true) value. The coefficients a and b are calculated using the following formulas,$$a=\frac{\sum \left(x_{i}-\bar{x})(y_{i}-\bar{y}\right)}{\sum {\left(x_{i}-\bar{x}\right)}^{2}},\;b=\bar{y}-a\bar{x},$$
where $x_{i}$ is the measured hemoglobin (Hb) value, $y_{i}$ is the actual hemoglobin concentration, and $\bar{x},\;\bar{y}$ denote their respective mean values.

### 3.3. Simulation Results

The performance metrics of the proposed analog front-end circuit, including the LED driver and the optoelectronic Rx, are summarized in Table 4. Key parameters such as gain, bandwidth, power consumption, and core layout area are evaluated through the post-layout simulations.

Table 5 presents a comparative analysis between the proposed digital signal processing modules and the existing commercial systems. Although the commercial pulse oximeters (POs) are not designed to measure the total hemoglobin (tHb) concentrations, the tHb values estimated by the proposed system are validated against the clinical blood test results. This comparative approach enables the indirect evaluation of tHb estimation accuracy in the absence of direct reference measurements from commercial devices.

## 4. Discussions

### 4.1. Algorithm Verification

Figure 16 illustrates the flowchart of algorithm validation, where two methods for estimating the DC component of the PPG signal are evaluated by using identical input data. Algorithm A employs the average value of the waveform, whereas Algorithm B uses the local minimum value. This quantitative comparison revealed that Algorithm B produces smaller errors in estimating the tHb and SpO_2_ values, and it was therefore selected for the final system implementation.

### 4.2. Analog Circuit Verification Using Virtuoso

The LED driver is implemented as a constant-current circuit to ensure stable operations across varying biological tissue conditions and to enable tunable optical intensity. Post-layout simulations under the worst-case PVT corner cases confirm that the output current variations across the three modulation modes remain within ±5%, thus verifying consistent drive capability under the process and temperature fluctuations.

The optoelectronic Rx circuit is also verified through the post-layout simulations using Cadence Virtuoso. The corresponding results confirm that the analog front-end achieves sufficient performance for PPG signal acquisition, including gain linearity, bandwidth limitation, and common-mode noise rejection.

### 4.3. Verification Using Programming Languages

To verify digital signal processing accuracy, the signal period of PPG calculated through fixed-point Verilog simulation on Xilinx Vivado 2023.1 is cross-checked against the results from Python 3.13.0 (Figure 17a) and MATLAB R2023a (Figure 17b). As Python indexing starts from zero, the dominant frequency index appears offset by one compared to MATLAB.

The heart rate calculated in MATLAB (with four-decimal precision) is considered the reference value. The proposed system’s output shows an error within an acceptable margin, therefore confirming the correctness of its frequency extraction algorithm.

### 4.4. Verilog HDL Simulation Verification

The entire signal processing pipeline is implemented by using Verilog HDL and is simulated by using Xilinx Vivado with Q2.23 fixed-point arithmetic. From the Vivado simulation results, the Hb and HbO_2_ concentrations are produced as 235,182 and 11,590,641, respectively, both scaled by 10^6^. This corresponds to the actual values of 0.235 g/dL (Hb) and 11.590 g/dL (HbO_2_). The calculated SpO_2_ value is 98%. These values fall within the normal physiological ranges and show over 95% agreement with the reference calculations performed in Python and MATLAB.

## 5. Conclusions

This paper presents a compact and fully integrated optical monitoring system for bio-signals that is capable of simultaneously measuring tHb, SpO_2_, and the heart rate in a non-invasive manner. The receiver incorporates an on-chip APD followed by an IA-based TIA, thus enabling high-gain and low-noise performance and reducing the system size and cost by eliminating the need for discrete photodetectors. Also, a five-wavelength LED source (670, 770, 810, 850, and 950 nm) is adopted to improve the spectral resolution between Hb and HbO_2_. Using this approach, we achieved a tHb estimation error of ±0.3 g/dL when compared against the invasive reference (blood test).

The LED driver includes a wide-swing cascode current mirror for flexible current modulations against the varying PVT conditions. The optoelectronic Rx employs a CMOS instrumentation amplifier with Miller compensation, yielding a transimpedance gain of 129 dBΩ and a bandwidth of 85 kHz while effectively suppressing high-frequency noise.

On the digital side, a fully parallel Verilog HDL architecture is implemented to perform band-pass filtering, FFT, AC/DC extraction, and the Modified Beer–Lambert Law (MBLL)-based quantification. Consequently, in Vivado simulations, the heart rate deviated by no more than ±0.2% from the MATLAB reference, the tHb exhibited an error of ±0.3 g/dL, and the SpO_2_ achieved ±1.5% accuracy, all with respect to their corresponding simulated references. Also, a valley detection algorithm is introduced to enhance robustness against motion artifacts.

As summarized in Table 6, the proposed CMOS hybrid system outperforms the existing solutions in terms of the measurement accuracy of SpO_2_, heart rate, and tHb, while maintaining moderate power consumption. Although the analog front-end IC design imposes inherent limitations in low-frequency filtering, these challenges can be effectively mitigated through FPGA-based digital signal processing. Hence, overall computational errors can be maintained below 1%, confirming the robustness and reliability of the proposed hybrid system. It is noted that the peak power consumption is higher than forcommercial products. However, the average power can be reduced significantly because the NMOS switch disconnects the LED current path whenever no drive pulse is applied and the digital modules employ clock gating to suppress dynamic power. Moreover, the device itself is intended to operate predominantly in a normal-power mode rather than high-power mode. In consequence, the time-averaged power consumption can be lower than the maximum power of 92.4 mW under typical operating conditions.

Given the increasing demand for continuous and non-invasive tHb monitoring in clinical, ambulatory, and athletic settings, the proposed CMOS hybrid system demonstrates a strong potential for commercialization. Its modular, hardware-based architecture is suitable for integration into wearable platforms such as smartwatches and patch-type biosensors.

## Figures and Tables

**Figure 1 micromachines-16-01086-f001:**
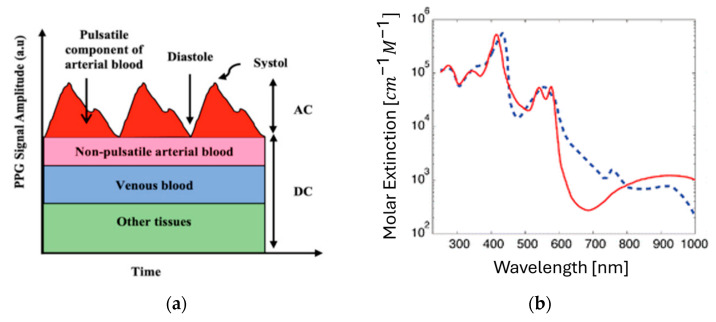
(**a**) AC and DC components of a PPG. (**b**) Absorption spectra of HbO_2_ and Hb.

**Figure 2 micromachines-16-01086-f002:**
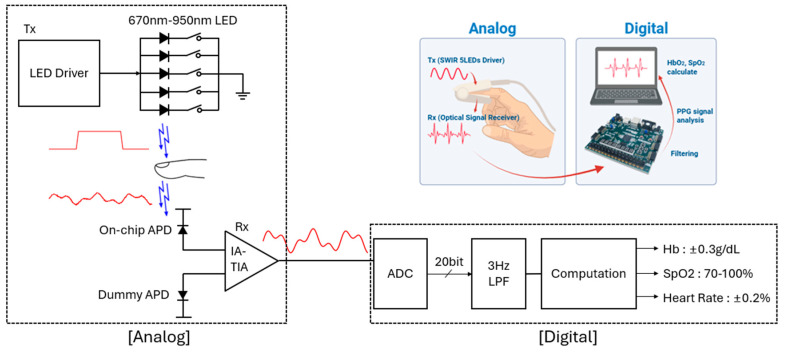
Block diagram of the proposed hybrid multi-wavelength PPG monitoring system.

**Figure 3 micromachines-16-01086-f003:**
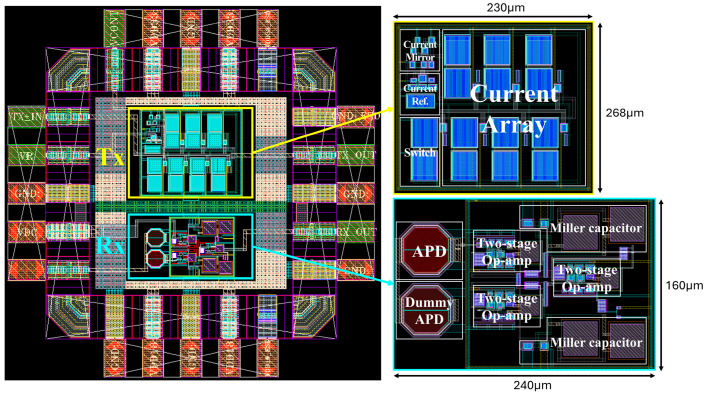
Layout of the analog front-end circuit including an LED driver and an optoelectronic Rx.

**Figure 4 micromachines-16-01086-f004:**
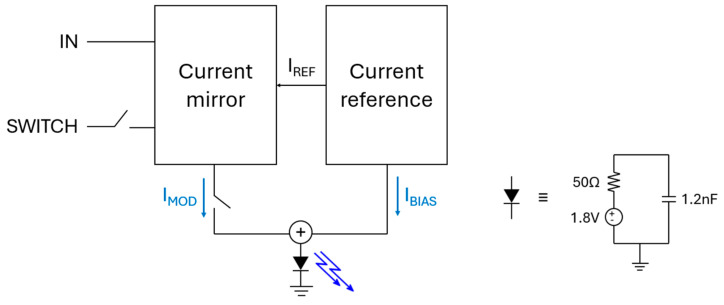
Block diagram of the proposed LED driver.

**Figure 5 micromachines-16-01086-f005:**
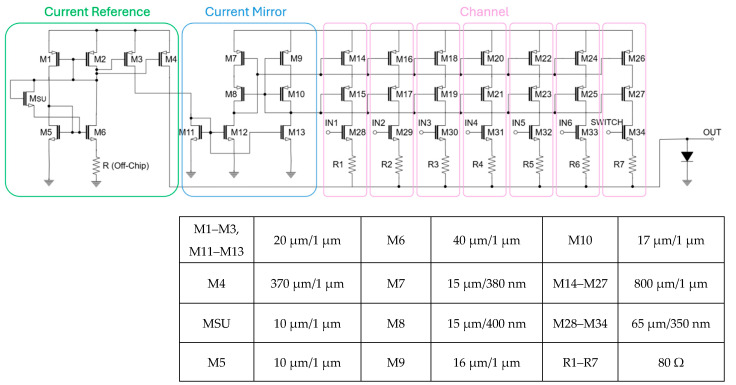
Schematic diagram of the proposed current-mode LED driver and its key parameters.

**Figure 6 micromachines-16-01086-f006:**
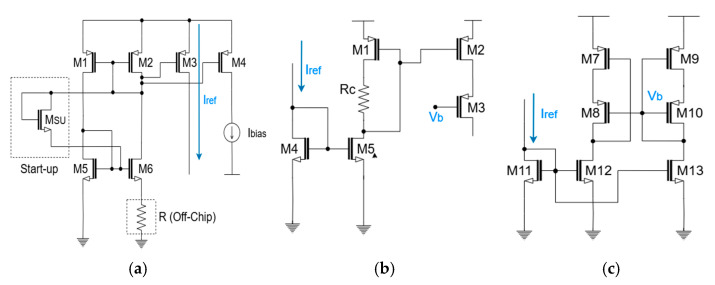
Various current mirrors: (**a**) reference current circuit with startup device, (**b**) conventional cascode current mirror, (**c**) wide-swing cascode current mirror.

**Figure 7 micromachines-16-01086-f007:**
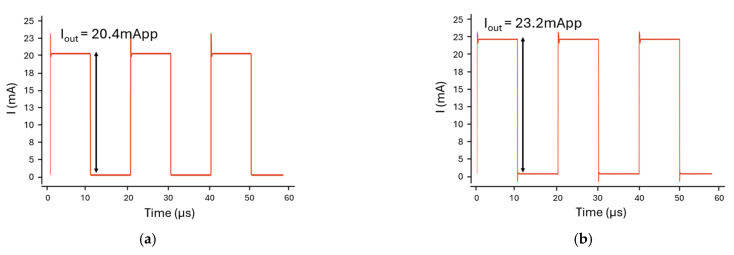
Output currents of the LED driver: (**a**) switch OFF, (**b**) switch ON.

**Figure 8 micromachines-16-01086-f008:**
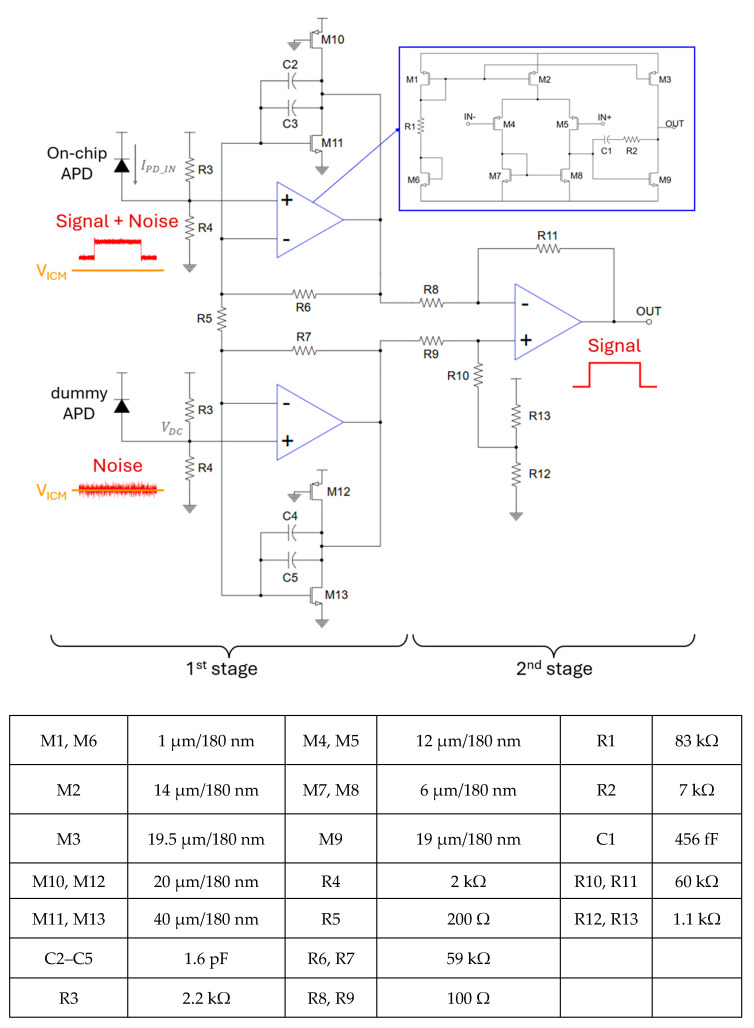
Schematic diagram of the proposed optoelectronic Rx (IA-TIA) and its key parameters.

**Figure 9 micromachines-16-01086-f009:**
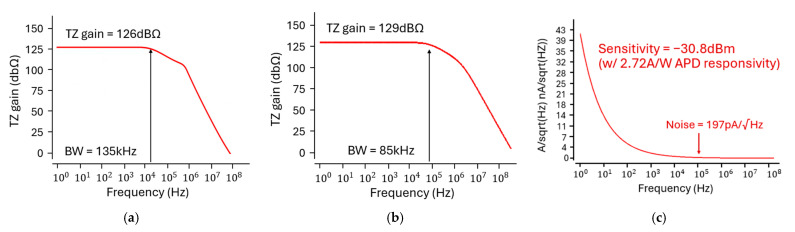
Frequency responses of (**a**) a conventional IA without Miller capacitor, (**b**) the proposed IA-TIA with Miller capacitor, and (**c**) its simulated noise response.

**Figure 10 micromachines-16-01086-f010:**
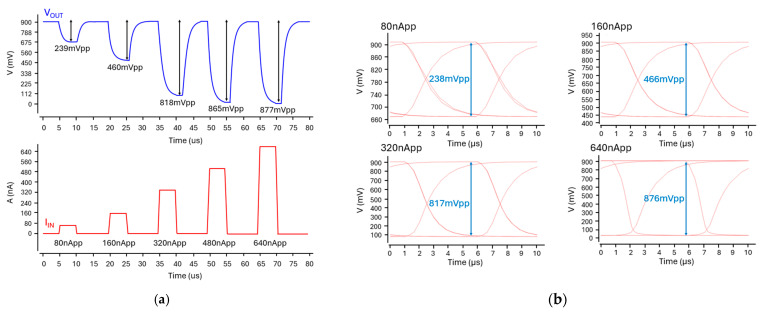
(**a**) Simulated pulse response and (**b**) simulated eye-diagrams at the data rate of 100 kb/s.

**Figure 11 micromachines-16-01086-f011:**
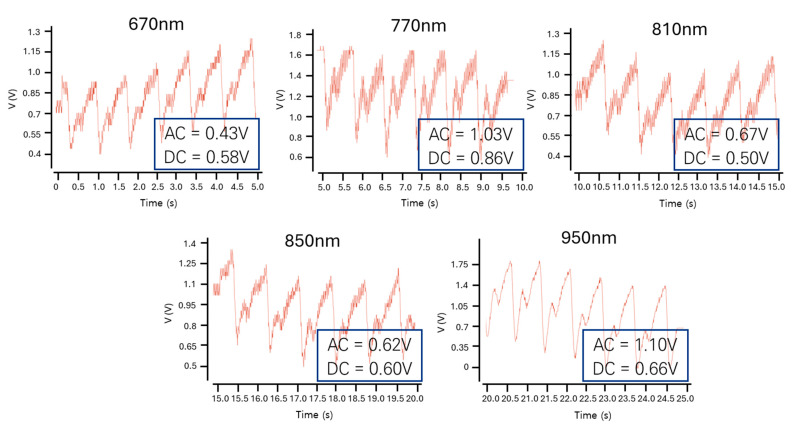
Transient responses of the optoelectronic Rx (IA-TIA) at each wavelength.

**Figure 12 micromachines-16-01086-f012:**
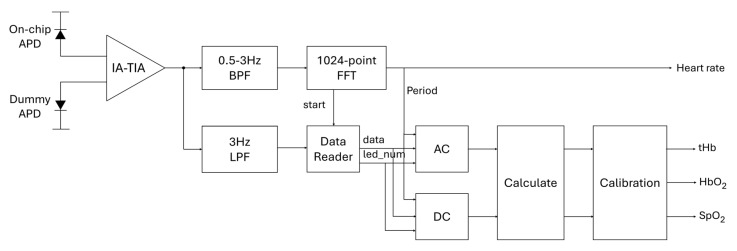
Signal flow diagram of the digital circuit.

**Figure 14 micromachines-16-01086-f014:**
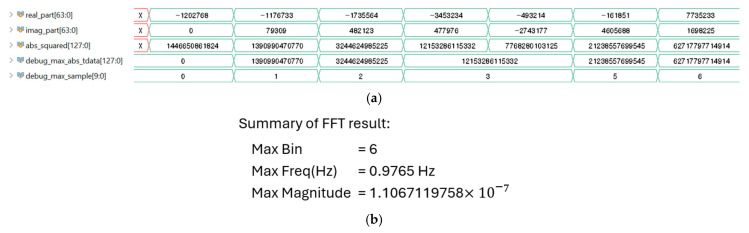
FFT simulation results: (**a**) maximum FFT magnitude, (**b**) maxbin index and FFT output.

**Figure 15 micromachines-16-01086-f015:**
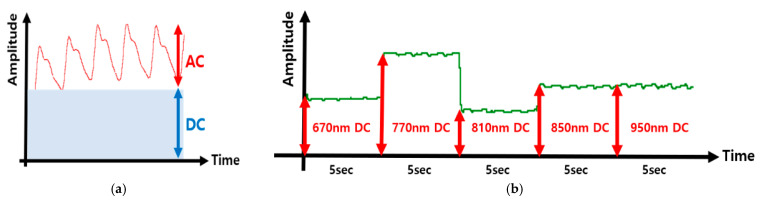
AC and DC modules based on Xilinx Vivado 2023.1 simulations: (**a**) AC and DC components of the PPG signal, (**b**) wavelength-specific DC components.

**Figure 16 micromachines-16-01086-f016:**
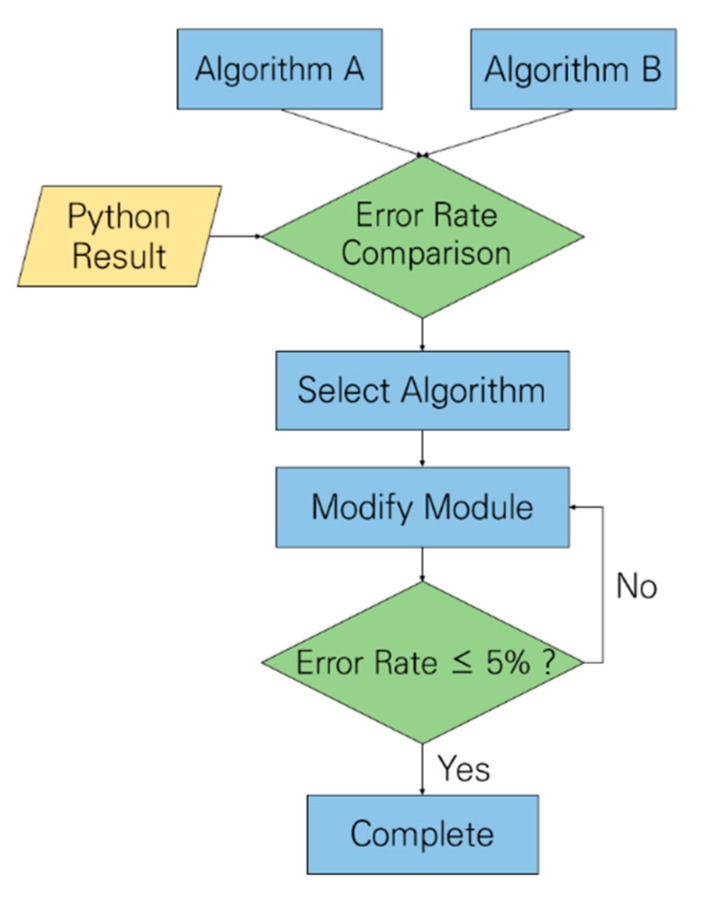
Validation workflow diagram.

**Figure 17 micromachines-16-01086-f017:**
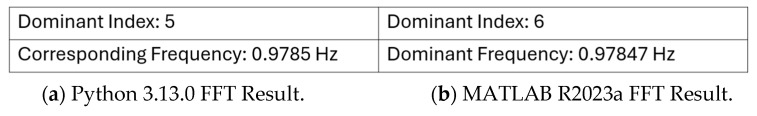
FFT results after band-pass filtering (BPF).

**Table 1 micromachines-16-01086-t001:** Definition and normal range of tHb and SpO_2_.

Term	Definition	Normal Range
Total Hemoglobin (tHb)	Concentration of all hemoglobin in the blood (oxygenated hemoglobin (HbO_2_) + deoxygenated hemoglobin (Hb))	Female: 12–16 g/dL Male: 14–18 g/dL
Blood Oxygen Saturation (SpO_2_)	Percentage of hemoglobin molecules bound with oxygen relative to the total hemoglobin SpO_2_ = (HbO_2_/tHb) × 100	95–100%

**Table 2 micromachines-16-01086-t002:** Comparison of HbO_2_ and Hb absorption with respect to wavelengths.

Wavelength	HbO_2_ Absorption	Hb Absorption	Characteristics
670 nm (Red)	Low	High	Hb-selective absorption
950 nm (NIR)	High	Low	HbO_2_-selective absorption
810 nm	Medium	Medium	Isosbestic point

**Table 3 micromachines-16-01086-t003:** Target performance of the proposed hybrid monitoring system.

Parameter	Target Performance	Benchmark/Remark
tHb concentration	±0.3 g/dL	Over 80% improvement vs. 3-wavelength PPG (avg. error 1.38 g/dL); comparable to blood test accuracy
SpO_2_ concentration	±1.5%	Comparable to commercial pulse oximeters
Heart rate	±0.2%	Higher precision than commercial pulse oximeters
Real-time and robustness	FPGA-based high-speed AC/DC extraction Motion artifact suppression via local peak filtering	Reliable under movement and noise

**Table 4 micromachines-16-01086-t004:** Post-layout simulation results of the analog circuits (Tx and Rx).

Parameters	LED Driver (Tx)		Optoelectronic Rx	
CMOS Process (nm)	180		180	
Operating Voltage (V)	3.3		1.8	
Optical Components	Five-wavelength multichip LED emitter		TIA integrated w/on-chip APD	
Signal Configuration	Single-ended		Single-ended	
Topology	Common cathode		IA-based TIA	
Performance	Modulation current (mA)	0–23.5	TIA gain (dBΩ)	129
	Bias current (mA)	3.6	Bandwidth (kHz)	85
Max Power Consumption (mW)	73.4		19	
Area (µm^2^)	230 × 268		240 × 160	

**Table 5 micromachines-16-01086-t005:** Digital module implementations.

Parameters	Proposed Method	Blood Test	Commercial Noninvasive Devices
tHb (g/dL)	11.8	11.5	Not provided
SpO_2_ (%)	98	–	98–99
Filter (Hz)	0.5–3	–	0.5–5.5
Heart Rate (bpm)	58.6	–	60

**Table 6 micromachines-16-01086-t006:** System-level performance comparison.

Parameter		Proposed Method	Commercial Non-Invasive Products
SpO_2_ (%)	Measurement Range	70–100	70–100
	Accuracy	Within ±1.5%	Within ±2%
Heart Rate (bpm)	Measurement Range	30–180	30–235
	Accuracy	Within ±0.2%	Within ±2%
tHb (g/dL)		±0.3	Not provided; blood test results used as target
Max Power Consumption (mW)		92.4	<40

## Data Availability

Data are contained within the article.

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
