# Peer review of "A CMOS Hybrid System for Non-Invasive Hemoglobin and Oxygen Saturation Monitoring with Super Wavelength Infrared Light Emitting Diodes"

_micromachines, 2025, doi:10.3390/mi16101086_

Round 1

Reviewer 1 Report

Comments and Suggestions for Authors

The manuscript presents a CMOS-based hybrid system capable of noninvasively quantifying the total hemoglobin (tHb), the oxygen saturation (SpOâ‚‚), and the heart rate by utilizing five-wavelength (670, 770, 810, 850, and 950 nm) photoplethysmography, achieving an accuracy of ±0.3 g/dL in tHb estimation, with an 80% improvement over conventional three-wavelength designs. Given its integrated modular, hardware-based architecture, the proposed CMOS hybrid system seems a potential commercialization for continuous and non-invasive tHb monitoring in clinical, ambulatory, and athletic settings.

This review sounds interesting as highlighting a new system for measuring the total hemoglobin, which is essential for assessing oxygen transport capacity in many clinic conditions. Overall, some minor points should be considered by the authors before it can be recommended for publication.

  1. In Page 2, line 56, “Compared to conventional three-wavelength PPG approaches, ……” The study used five wavelengths (670, 770, 810, 850, and 950 nm) instead of conventional three-wavelength for the PPG-based measurement. So, how do the two wave lengths of 770 and 850nm added greatly reduce the measurement error? Should be more clarified for a better understanding.
  2. In Figure 7, what does it mean of VICM and Vlight? Should be clarified more in the context for a better readership.
  3. Finally, the proposed method presents a power consumption of 92.4 mW, compared to the commercial one less than 40 mw. I think the power consumption is a little bit of higher than the commercia non-invasive products. More clarification should be given for this shortage in Discussions.

Author Response

1. In Page 2, line 56, “Compared to conventional three-wavelength PPG approaches, ……” The study used five wavelengths (670, 770, 810, 850, and 950 nm) instead of conventional three-wavelength for the PPG-based measurement. So, how do the two wavelengths of 770 and 850nm added greatly reduce the measurement error? Should be more clarified for a better understanding.

--> (ans.) The addition of two wavelengths is not merely an incremental increase in data points. It is intended to fundamentally transform the measurement model into a robust overdetermined system and to more precisely correct for the effects of tissue scattering.

A three-wavelength approach typically forms a determined system, where three unknown variables (e.g., concentrations of oxyhemoglobin (HbOâ‚‚), deoxyhemoglobin (Hb), and a tissue scattering coefficient) are solved by using exactly three measurements (and/or equations). While theoretically solvable, this system is highly susceptible to the subtle noise inherent in biological signals. If a minor error occurs in any single wavelength, there is no redundant information to compensate for it, thus leading to the amplification of this error in the final calculated concentrations.

In contrast, the five-wavelength method constitutes an overdetermined system by utilizing five equations to solve for the same unknowns. This allows for the use of the least-squares method to find an optimal solution that minimizes the overall error across all five measurements. In this process, random noise from any single channel can be effectively averaged out and compensated for by the data from the other channels. This fundamental shift in the mathematical model is the key principle that secures the system's stability and robustness against noise, thereby dramatically reducing measurement error. We have added this information to Section 2 (lines 80-141).

2. In Figure 7, what does it mean of VICM and Vlight? Should be clarified more in the context for a better readership.

--> (ans.) Figure 7 in the original manuscript now becomes ‘Figure 8’ in the revised version.

(1) For a better readership, we have removed Vlight from the figure.

(2) VICM denotes the common-mode voltage occurring at the two input nodes by the input resistors R3 and R4. Therefore, this VICM is common for both the on-chip APD channel and the dummy APD channel. In the former channel, an offset current is generated from the on-chip APD. This offset current is converted to a DC offset voltage, hence making the input pulse float over VICM, as shown at the input of the former channel. Meanwhile, the latter channel with a shielded dummy APD is optically blocked and therefore generates only common-mode noises.

We have added this information to Section 3.1 on page 8 (lines 261-266).

3. Finally, the proposed method presents a power consumption of 92.4 mW, compared to the commercial one less than 40 mW. I think the power consumption is a little bit of higher than the commercia non-invasive products. More clarification should be given for this shortage in Discussions.

--> (ans.) The transmitter is the dominant contributor to our power budget, so while the reported maximum power is higher than that of some commercial devices, the average consumption can be reduced. Specifically, an NMOS switch in the transmitter disconnects the LED current path when no input pulse is applied, and the digital modules employ clock-gating. In addition, the device is operated predominantly in the normal-power mode rather than the high-power mode. Consequently, the time-averaged power is expected to be lower than 92.4 mW. We have added this information to Section 5 on page 17 (lines 512-518).

Reviewer 2 Report

Comments and Suggestions for Authors

This paper focuses on a multiwavelength LED-based system to non-invasively monitor total hemoglobin, oxygen saturation and hear rate. Much of the paper is devoted to the details of the circuit design for a 5-channel LED driver, the detection circuit and the signal processing methods. Validation of the performance of the system is undertaken via simulation.

The proposed system appears to be novel, and may be of interest to readers. However, at present the paper is not publishable since it does not directly cite any of the references in the text. I assume that was due to some sort of unfortunate formatting issue but it does make the work of the reviewer more difficult since we are left to guess where in the text the articles were cited.

Even assuming that the citations were included properly, there are two other aspects that should be addressed. The first is that I do not see any other discussion of previously published multichannel optical detection of these same parameters, even though several have been published in previous years. Secondly, the abstract leads the reader to assume that the impressive levels of precision that were reported were achieved in clinical samples, or at least in some of physical implementation. However, it then becomes apparent by the end of the paper that this was all undertaken in simulation.

Other comments:

In Fig. 4, all channels appear to be connected to a single LED, whereas I assume that each switched channel should address its own LED, no?

Fig 10. should make clear in the caption that this is simulated data 

On page 4, PVT is used, but only defined later

I would probably have more comments if and when I see a version with citations.

Author Response

1. The proposed system appears to be novel, and may be of interest to readers. However, at present the paper is not publishable since it does not directly cite any of the references in the text. I assume that was due to some sort of unfortunate formatting issue but it does make the work of the reviewer more difficult since we are left to guess where in the text the articles were cited.

--> (ans.) Several relevant references have now been explicitly cited in the main text, and the revised manuscript highlights these additions for clarity.

2. Even assuming that the citations were included properly, there are two other aspects that should be addressed. The first is that I do not see any other discussion of previously published multichannel optical detection of these same parameters, even though several have been published in previous years.

--> (ans.) In response to this comment, we have added a new section (Section 2) to discuss prior studies. In this section, we explain that multi-wavelength systems can reduce measurement error compared to three-wavelength systems when measuring the same parameters.

Furthermore, to differentiate our work from other five-wavelength systems, we have clarified that our proposed system is a CMOS hybrid architecture that utilizes both an analog front-end and a digital module, highlighting its distinction from previous research.

3. Secondly, the abstract leads the reader to assume that the impressive levels of precision that were reported were achieved in clinical samples, or at least in some of physical implementation. However, it then becomes apparent by the end of the paper that this was all undertaken in simulation.

--> (ans.) We have revised the abstract to clarify that the system's reported accuracy is based on simulation results (Page 1, lines 22–23).

However, we have also noted that the input parameters for the simulations were derived from experimental measurements on a human finger. Furthermore, the accuracy for total hemoglobin concentration was determined by comparing the simulation results with values from invasive blood tests.

4. In Fig. 4, all channels appear to be connected to a single LED, whereas I assume that each switched channel should address its own LED, no?

--> (ans.) Figure 4 in the original manuscript now becomes Figure 5 in the revised version.

We have revised the description of the LED driver to clarify that the six channels jointly drive a single multi-chip LED emitter, which integrates five wavelengths, i.e., 670, 770, 810, 850, and 950 nm (Page 7, lines 225–233).

To further clarify, the LEDs for the five wavelengths are not activated simultaneously. Instead, they are activated sequentially through switching. Therefore, it is sufficient to drive only one LED at a time for the system's operations.

5. Fig. 10 should make clear in the caption that this is simulated data.

--> (ans.) We have revised Figure 13 (formerly Figure 10 in the submitted manuscript) and updated its caption to clearly and intuitively show that it presents simulated data.

6. On page 4, PVT is used, but only defined later.

--> (ans.) We have revised the manuscript (lines 198-199).

Round 2

Reviewer 2 Report

Comments and Suggestions for Authors

The authors have addressed my comments and concerns, and the paper is suitable for publication now. I appreciate the efforts they made to explain the scientific basis of their approach.